# Clinical outcome prediction from analysis of microelectrode recordings using deep learning in subthalamic deep brain stimulation for Parkinson's disease

Kwang Hyon Park[1°], Sukkyu Sun[2°], Yong Hoon Lim[1], Hye Ran Park[3], Jae Meen Lee[4], Kawngwoo Park[5], Beomseok Jeon[6], Hee-Pyoung Park[7], Hee Chan Kim[2,8,9‡]*, Sun Ha Paek[1,10,11‡]*

1 Department of Neurosurgery, Seoul National University Hospital, Seoul, Korea, 2 Interdisciplinary Program in Bioengineering, Graduate School, Seoul National University, Seoul, Korea, 3 Department of Neurosurgery, Soonchunhyang University Seoul Hospital, Seoul, Korea, 4 Department of Neurosurgery, Pusan National University Hospital, Busan, Korea, 5 Department of Neurosurgery, Gachon University Gil Medical Center, Incheon, Korea, 6 Department of Neurology, Seoul National University Hospital, Seoul, Korea, 7 Department of Anesthesiology and Pain Medicine, Seoul National University Hospital, Seoul National University College of Medicine, Seoul, Korea, 8 Department of Biomedical Engineering College of Medicine, Seoul National University, Seoul, Korea, 9 Institute of Medical & Biological Engineering, Medical Research Center, Seoul National University, Seoul, Korea, 10 Ischemia Hypoxia Disease Institute, Seoul National University College of Medicine, Seoul, Korea, 11 Cancer Research Institute, Seoul National University College of Medicine, Seoul, Korea

° These authors contributed equally to this work.
‡ These authors also contributed equally to this work.
* paeksh@snu.ac.kr (SHP); hckim@snu.ac.kr (HCK)

## Abstract

### Background

Deep brain stimulation (DBS) of the subthalamic nucleus (STN) is an effective treatment for improving the motor symptoms of advanced Parkinson's disease (PD). Accurate positioning of the stimulation electrodes is necessary for better clinical outcomes.

### Objective

We applied deep learning techniques to microelectrode recording (MER) signals to better predict motor function improvement, represented by the UPDRS part III scores, after bilateral STN DBS in patients with advanced PD. If we find the optimal stimulation point with MER by deep learning, we can improve the clinical outcome of STN DBS even under restrictions such as general anesthesia or non-cooperation of the patients.

### Methods

In total, 696 4-second left-side MER segments from 34 patients with advanced PD who underwent bilateral STN DBS surgery under general anesthesia were included. We transformed the original signal into three wavelets of 1–50 Hz, 50–500 Hz, and 500–5,000 Hz. The wavelet-transformed MER was used for input data of the deep learning. The patients were divided into two groups, good response and moderate response groups, according to

**Data Availability Statement:** All relevant data are within the paper and its Supporting Information files.

**Funding:** This study was supported by the Korea Healthcare Technology R&D Project (grant No. HI11C21100200), funded by the Ministry of Health & Welfare, Republic of Korea; the Industrial Strategic Technology Development Program (grant No. 10050154, Business Model Development for Personalized Medicine Based on Integrated Genome and Clinical Information) funded by the Ministry of Trade, Industry & Energy (MI, Korea); the Original Technology Research Program for Brain Science through the National Research Foundation of Korea (NRF) funded by the Ministry of Education, Science and Technology (grant No. 2015M3C7A1028926); the Original Technology Research Program for Brain Science through the National Research Foundation of Korea (NRF) funded by the Ministry of Education, Science and Technology (grant No. 2017M3C7A1047392); and by Soonchunhyang University research funds.

**Competing interests:** The authors have declared that no competing interests exist.

DBS on to off ratio of UPDRS part III score for the off-medication state, 6 months postoperatively. The ratio were used for output data in deep learning. The Visual Geometry Group (VGG)-16 model with a multitask learning algorithm was used to estimate the bilateral effect of DBS. Different ratios of the loss function in the task-specific layer were applied considering that DBS affects both sides differently.

## Results

When we divided the MER signals according to the frequency, the maximal accuracy was higher in the 50–500 Hz group than in the 1–50 Hz and 500–5,000 Hz groups. In addition, when the multitask learning method was applied, the stability of the model was improved in comparison with single task learning. The maximal accuracy (80.21%) occurred when the right-to-left loss ratio was 5:1 or 6:1. The area under the curve (AUC) was 0.88 in the receiver operating characteristic (ROC) curve.

## Conclusion

Clinical improvements in PD patients who underwent bilateral STN DBS could be predicted based on a multitask deep learning-based MER analysis.

## Introduction

Deep brain stimulation (DBS) of the subthalamic nucleus (STN) is an effective way to improve motor complications in advanced Parkinson disease (PD) [1]. The definition and targeting of the STN has been improved due to advances in magnetic resonance imaging (MRI) resolution [2]. While brain shifting and intrinsic inaccuracies in frame-based targeting are still obstacles, intraoperative signal analyses from microelectrode recording (MER) are usually performed [3]. Although papers have classified and analyzed MER signals and correlated them with clinical outcomes [4–6], no studies have analyzed MER signals using deep learning with the clinical outcomes of patients with advanced PD after STN DBS.

Deep learning is a machine learning method that allows for learning features of input data. Among the machine learning methods, convolutional neural network (CNN) models can process signals using imaging processing methods. In medical applications, CNN is used for analyzing medical data such as X-ray, MRI, CT images and fundoscopy [7–10]. In signal processing, CNN methods are used for several cases [11, 12]. Converting a signal to an image with wavelet transformation is an effective way to extract features. High-frequency signals can be reduced to a single size of the image regardless of the size of the original signal.

In this study, we analyzed outcome-guided deep learning-based prediction of motor function improvement after bilateral STN DBS in PD patients. We transformed MER signals to the wavelet domain, which has a time axis, and compared them with the Unified Parkinson's Disease Rating Scale (UPDRS) part III scores of patients with advanced PD at 6 months after STN DBS. We investigated the ranges of the MER signal bands and the optimal deep learning algorithm, by which the clinical outcome of motor function improvement is better predicted in patients with advanced PD after bilateral STN DBS. The ability to recommend the optimal lead insertion location through MER signals in the operating room would be of great help to less experienced neurosurgeons. We could improve the outcome even under restrictions such as general anesthesia or noncooperation of the patients. In addition, discovering new findings that have not been found by conventional waveform analysis methods may be possible.

## Methods

### Subjects

Forty-four patients with advanced PD who underwent bilateral STN DBS with MER under general anesthesia between 2014 and 2017 were included in this study. Six patients who had missing clinical data and four patients who had two active contacts of an electrode at 6 months after STN DBS were excluded. The clinical information of all patients was retrospectively reviewed. The UPDRS scores of every patient were evaluated preoperatively and at 6 months after surgery by experienced neurologists. The clinical evaluation of the UPDRS scores was performed under two conditions: off-medication (when the patients had taken no medications for 8 to 12 hours) and on-medication (when the patients had experienced maximal clinical benefit 1 to 3 hours after the usual morning dose of a dopaminergic treatment) before and at 6 months after STN DBS. However, in this study only the off-medication status was included. MER was performed at 1-mm intervals from 10 mm to 5 mm above the target and at 0.5-mm intervals from 5 mm above to 5 mm below the target. As the lead contacts are 1.5 mm long and MER was performed at intervals of 0.5 mm, the three MER signals were inevitably matched to the same clinical outcome as the input and output values in the deep learning model. When the lead stimulation depth contact was changed at 6 months after surgery, the MER signal of the depth thought to have been stimulated was selected.

### Surgical procedure

The surgical procedure was similar to that reported in previous studies [13, 14]. Anti-Parkinsonian medications were not stopped preoperatively. A stereotactic Leksell G frame (Elekta) was mounted on the head under local anesthesia. A 1.5 T brain MRI was performed (General Electric Medical Systems). The 3-D fast spoiled gradient-echo (FSPGR) sequence was used for anterior commissure (AC)–posterior commissure (PC) calculations. T2 spin-echo images were obtained to define the boundaries of the STN. SurgiPlan (Elekta) was used to run the simulations for targeting the sensorimotor region of the STN and selecting the trajectories. MERs were performed by means of a Leadpoint system (Medtronic) under general anesthesia. The depth of sedation was monitored by the bispectral index under total intravenous anesthesia with propofol and remifentanil. The propofol concentration was titrated to maintain the bispectral index value of 60–80. Permanent model 3389 (Medtronic) quadripolar electrodes were implanted along the proper trajectory to stimulate more sensorimotor regions of the STN, which was localized by both preoperative brain MRI and intraoperative MER. Left DBS was performed first, followed by right DBS. Programmable pulse generators (Medtronic) were implanted in the subclavicular region and connected to the electrodes.

### Microelectrode recordings

We obtained the signal using a five-array microdrive. Each MER signal was filtered at 0.2–5,000 Hz (4 seconds). The signal was recorded through the Leadpoint output board (Medtronic) with a sampling rate of 48 KHz (gain: 1,000). The signal was stored using Spike2 (Cambridge Electronic Design Limited).

The MER signals for the deep learning algorithm were selected from the determined final DBS trajectory. Finally, the recorded MER signal was selected from the active contact point at 6 months after DBS. As the contact surface span was 1.5 mm and MER was performed at intervals of 0.5 mm, the three MER signals were inevitably matched to one clinical outcome value in the deep learning algorithm. We analyzed only the left MER signals as input data for the deep CNN. In total, 696 MER segments from 34 PD patients who underwent bilateral STN DBS

surgery under general anesthesia were included in this study. The datasets of thirty patients were assigned to the training set, and those of four patients were assigned to the test set.

## Wavelet transformation

Several ways to visualize signals, such as using a short-time Fourier transform (STFT) and a wavelet transform, are possible [15]. By visualizing the signal, one can see which signal is dominant in the frequency domain. The wavelet transform is a well-known approach in time-frequency analyses such as MER. It was originally developed for seismic signal processing [16], but it is also used for neural spike-train composition [17]. Compared with Fourier transformations, wavelet transformations can reach the optimal time-frequency resolution due to an unfixed window size [18]. In general, a recurrent neural network (RNN) can analyze sequential or time series signals. However, using an RNN is difficult due to the lack of memory in the cell to analyze MERs because the sampling rate is high (sampling rate of a few kHz). In a wavelet analysis, a large sampling rate signal can be reduced to a few kilobytes of signal. We transformed the original signal into three wavelets of 1–50 Hz, 50–500 Hz, and 500–5,000 Hz. Finally, a 50~500 Hz wavelet with a time and frequency resolution of 8 was used. A continuous wavelet transform of signal $s(t)$ is defined by

$$CWT_{(a,b)}\{s(t)\} = \frac{1}{\sqrt{|a|}} \int_{-\infty}^{\infty} s(t) \Psi^* \left( \frac{t-b}{a} \right) dt$$

(T: time, a: scale parameter, b: translational parameter, $\Psi$: analyzing wavelet, and $*$: conjugate function) [19].

The scale parameter a is inversely proportional to frequency $f$. A Morlet wavelet is

$$\Psi(t) = \pi^{\frac{-1}{4}} e^{(i\omega_0 t)} e^{\left( \frac{-t^2}{2} \right)}$$

All signals were normalized to 0~1, and 60-Hz line noise was not filtered when transforming 50~500 Hz. The 60-Hz line noise has a very small area in whole WT. Therefore, the length of the wavelet was determined as 4 seconds considering the time-frequency resolution, the learning results and the MER segment number. The Morlet function was adopted as the mother wavelet function.

## Training set and test set

In total, 696 MER segments from 34 patients, with an average of 20.47 segments per person, were included in the analysis. The data set was shuffled to prevent one patient from including both the training set and test set. For the test set, 2 patients in the good-response group and 2 patients from the moderate-response group were randomly selected. According to the ratio of the DBS on/off score at 6 months after surgery, the patients were divided into two response groups: good and moderate. The UPDRS score of the right side is better than that of the left side, and the classification criteria are different for each side. For the left side, 0–0.69 is a good response, and 0.7–1.0 is a moderate response. For the right side, 0–0.59 is a good response, and 0.6–1.0 is a moderate response.

## Deep learning

Deep learning is a state-of-the-art approach that can extract both high-level and low-level abstractions of data. Well-known deep learning algorithms include CNNs, RNNs, and generative adversarial networks (GANs). Among these algorithms, CNNs are better for image classification. CNN extracts features from images using convolutional and pooling layers, and then

the extracted features are classified with a fully connected layer. One of the CNN structures, Visual Geometry Group 16 (VGG16) [20], has been widely used since the 2014 ImageNet Large Scale Visual Recognition Challenge (ILSVRC). VGG16 was used as the classification network. The fully connected layers were modified with 4,096, 4,096, and 1,024 classes to 120, 120, 16, and 2 classes with a dropout rate of 0.2. A learning rate of 0.001 was used for both multitask learning and labeling, and a gradient descent optimizer was used.

## Multitask learning

Multitask learning has been successfully used in machine learning and recently in computer vision [21]. Three types of multitask learning exist: joint learning, learning to learn, and learning with an auxiliary task. Joint learning was adopted. With multitask learning, a CNN was trained via two different labels. A model that learns two different tasks simultaneously can learn a more general representation [22]. One label is a regularizer. Learning with two labels is more generalizable than training with one label. The convolution layers of VGG16 were used as a shared layer, and they were used to train a fully connected layer separately with each side label.

We used the ipsilateral clinical outcome and contralateral clinical outcome as a label for each of the task-specific layers. Furthermore, we jointly trained a single network with two different labels with different ratios for the loss function because the effect of left-sided DBS on the contralateral motor function is consider greater than that on ipsilateral motor function. The algorithm learned with two labels. In the test set, however, the goal is to distinguish the good-response group and the moderate-response group of only the contralateral side. Fig 1 shows the whole process in a schematic diagram.

## Statistical analysis

Statistical analyses were performed with R 3.5.1. Pearson's correlation coefficients were calculated and are denoted by r.

## IRB

This retrospective study adhered to the Declaration of Helsinki. The patient records were anonymized and deidentified prior to the analysis. Patients provided their written informed

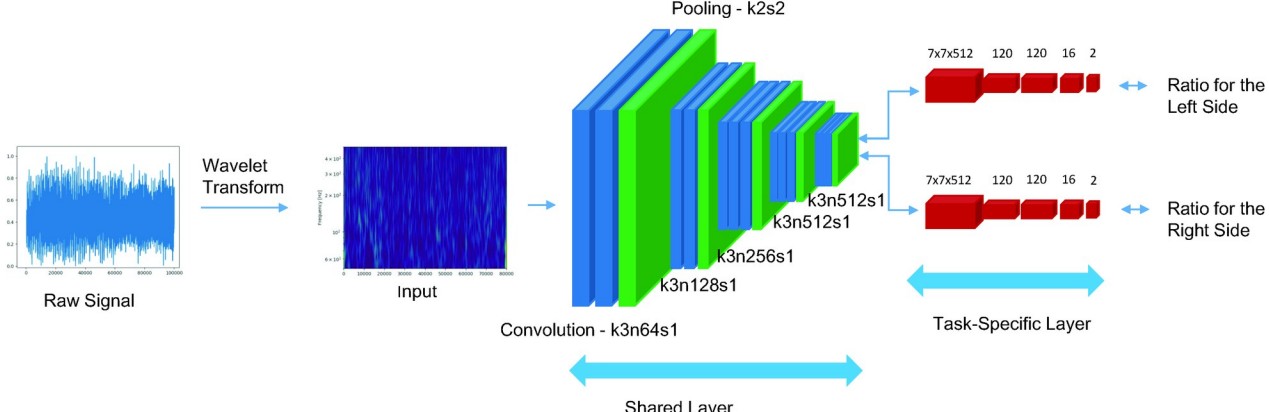

**Fig 1. Process schematic diagram.** The input of the convolutional neural network is the wavelet-transformed microelectrode recording signal. The output is the ratio of the preoperative and 6 months postoperative Unified Parkinson's Disease Rating Scale (UPDRS) part III score for each side in multitask learning. (k: kernel size, n: number of feature maps, s: strides).

consent to have their health information recorded and used for research purposes. The study protocol was approved by the Institutional Review Board of Seoul National University Hospital (IRB No. 1812-086-995).

## Results

### Patient data

The MER and clinical data of 34 patients were included in the study (Table 1). The mean age of the patients at diagnosis was 43.21 ± 7.96 years old, and the mean age of the patients at surgery was 57.21 ± 7.96 years old. The duration from symptom onset to surgery was 14.00 ± 5.62. Among the UPDRS part III scores, the subtotals of the items with left or right segments are denoted by "Left" or "Right", respectively. The mean preoperative UPDRS part III score was 12.41 ± 5.19 for the right side of the body and 12.76 ± 5.39 for the left side of the body. The mean off-DBS UPDRS part III score at 6 months after surgery was 11.22 ± 4.35 for the right side of the body and 11.54 ± 4.39 for the left side of the body. The mean on-DBS UPDRS part III score at 6 months after surgery was 6.81 ± 4.19 for the right side of the body and 7.62 ± 4.13 for the left of the body.

The preoperative UPDRS part III scores of the left and right sides of the body for the 34 patients showed a linear correlation (r = 0.77, p<0.001) (Fig 2). At 6 months after the surgery, this correlation also existed in both the DBS on (r = 0.79, p<0.001) and off (r = 0.74, p<0.001) states (Fig 3A and 3B). The preoperative UPDRS part III score did not affect the DBS on/off ratio at 6 months after the operation (Fig 4A and 4B). However, at 6 months the DBS on/off ratios of the left and right sides of the body were significantly correlated (r = 0.67, p<0.001) (Fig 5).

### Relationship between the MERs & clinical outcome

The MER data were passed through a 50–500 Hz bandpass filter, cut into 4-second lengths, normalized, and transformed into wavelets (Fig 6A to 6F). The MER data were then classified according to clinical outcome.

### Single-task learning and multitask learning

The MER data were divided according to the frequency band and transformed into wavelets; Maximum accuracies are obtained when the training softmax cross entropy loss is nearly zero

**Table 1. Off-medication scores in 34 patients with bilateral stimulation of the subthalamic nucleus.**

| Score | Score Before STN DBS | Score at 6 Months After STN DBS, Off-stimulation | Score at 6 Months After STN DBS, On-stimulation |
|---|---|---|---|
| UPDRS Part I | 3.94 ± 3.27 | NA | 3.12 ± 2.14 |
| UPDRS Part II | 23.76 ± 9.95 | NA | 14.09 ± 9.04 |
| UPDRS Part III Right | 12.41 ± 5.19 | 11.22 ± 4.35 | 6.81 ± 4.19 |
| UPDRS Part III Left | 12.76 ± 5.39 | 11.54 ± 4.39 | 7.62 ± 4.13 |
| UPDRS Part III Total | 40.29 ± 15.04 | 34.85 ± 12.25 | 22.26 ± 11.81 |
| UPDRS Total | 68.00 ± 22.98 | NA | 39.47 ± 17.85 |
| Global Stage of the Disease (Hoehn and Yahr) | 3.16 ± 0.89 | 2.81 ± 0.77 | 2.50 ± 0.65 |
| Global Activities of Daily Living (Schwab and England) | 45.59 ± 23.77 | NA | 64.12 ± 25.75 |
| Levodopa Equivalent Daily Doses | 1411.65 ± 547.32 | NA | 431.47 ± 286.67 |

The UPDRS Part III scores were improved significantly in the on-stimulation state.

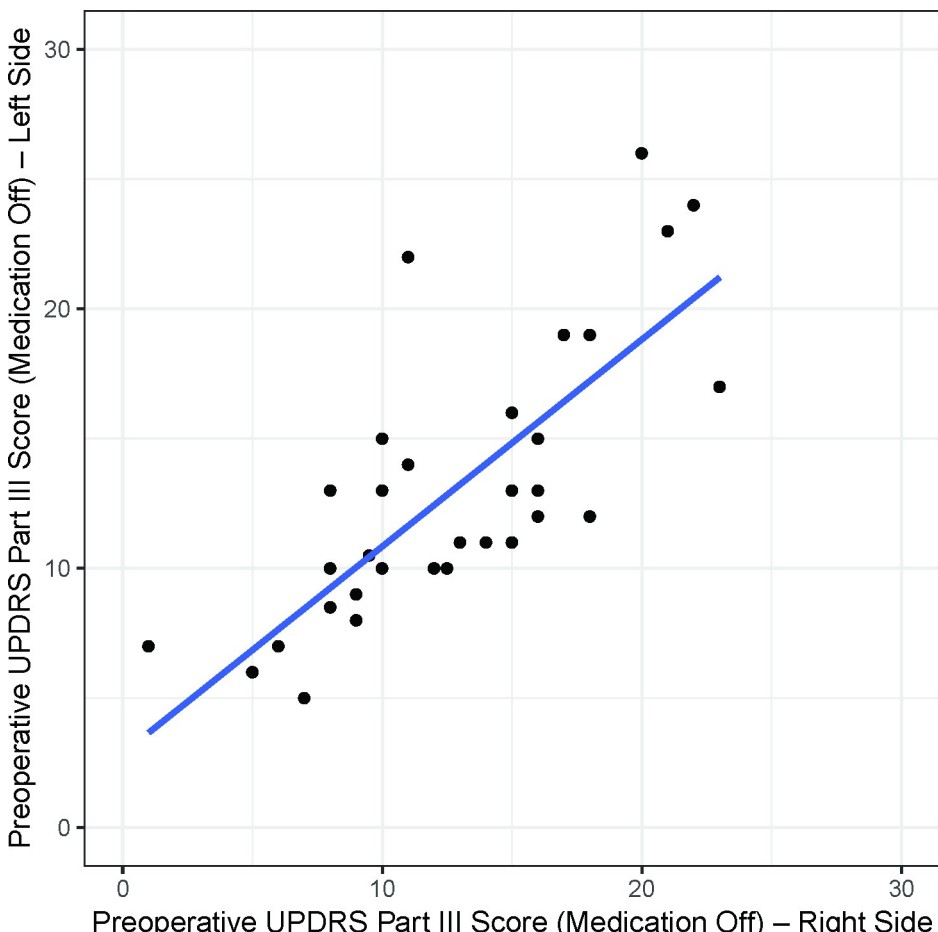

**Fig 2. The preoperative UPDRS part III scores (Medication Off).** The preoperative UPDRS part III scores (Medication Off) show linear correlations (r = 0.77, p<0.001) between the right and left sides of the body.

and the testing accuracy is highest (after epoch 50). The maximal accuracy was the highest in the 50–500 Hz group (75.0%) compared with the 1–50 Hz group (33.0%) and the 500–5,000 Hz group (62.5%). However, the model repeated learning for each epoch, and a fluctuation occurred in validation accuracy at the end of training (Fig 7A). Multitask learning was used to consider the effect of DBS on both sides of the body, and it found the ratio of maximal accuracy to the ratio of the left and right loss functions. The maximal accuracy (80.21%) occurred when the right-to-left loss ratio was 5:1 and 6:1 (77.08% at 4:1, 71.88% at 3:1, 65.63% at 2:1) (Table 2). The stability of the model with multitask learning was substantially improved over the model stability with single-task learning (Fig 7B). The receiver operating characteristic (ROC) curve was obtained from the prediction of the clinical outcome (Fig 8). The area under the curve (AUC) was 0.88.

## Discussion

### Single-task learning

When the wavelet transform of the entire MER is performed, it is divided into three frequency zones according to the resolutions available as the input of the CNN. These frequency zones are the same as the frequency bandpass ranges because the wavelet was divided according to

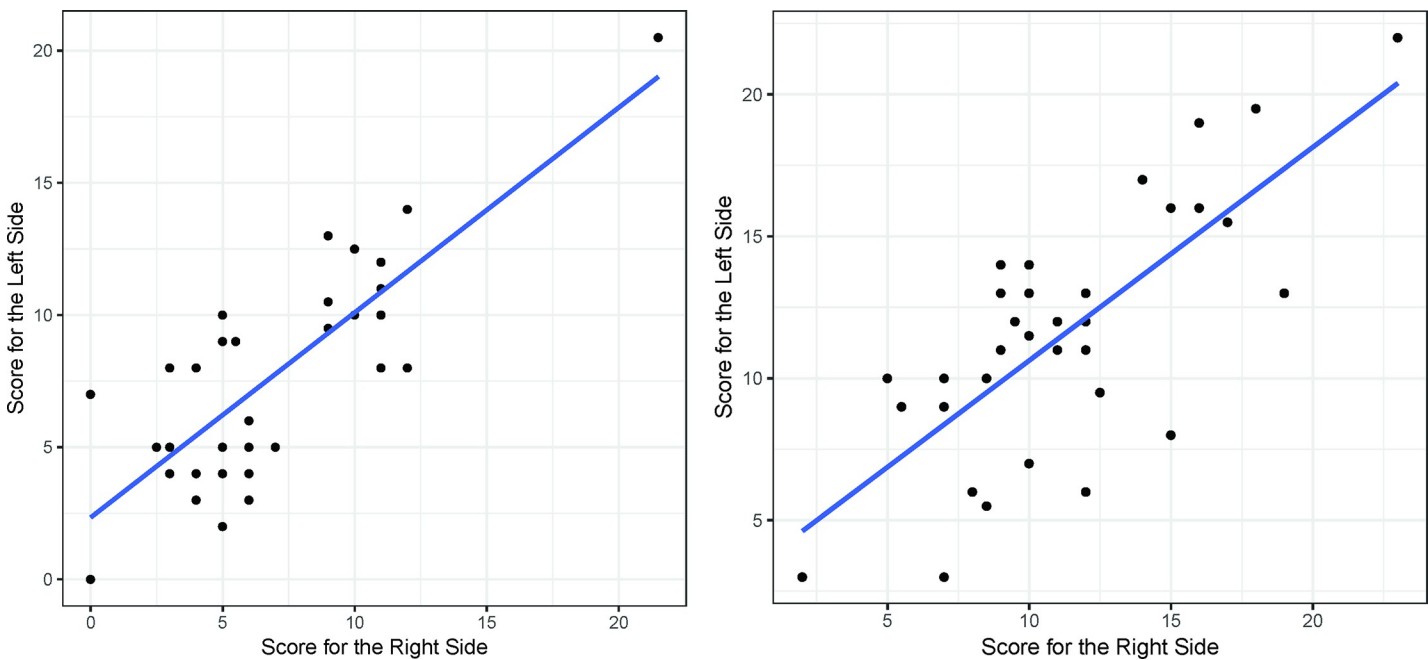

**Fig 3. The UPDRS part III scores at 6 months.** At 6 months after the surgery, the UPDRS Part III Scores show linear correlation (DBS On, Medication Off, r = 0.79, p<0.001)(A), (DBS Off, Medication Off, r = 0.74, p<0.001)(B).

frequency. Artifacts were the most common in the 1–50 Hz wavelet. In the 500–5,000 Hz wavelet, the spike appeared to be the most noticeable, but it was also observed in the 50–500 Hz wavelet. Typical firing patterns, particularly asymmetrical spikes with bursting patterns at high frequencies, and proprioceptive responses to passive movements of the joint have been used to determine the boundary of the STN [1]. However, because $\beta$-oscillatory activity (13–30 Hz) is limited to the dorsolateral oscillatory region of the STN, the length of the dorsolateral oscillatory region recorded in the macroelectrode-implanted trajectory predicted a favorable response to STN DBS [5].

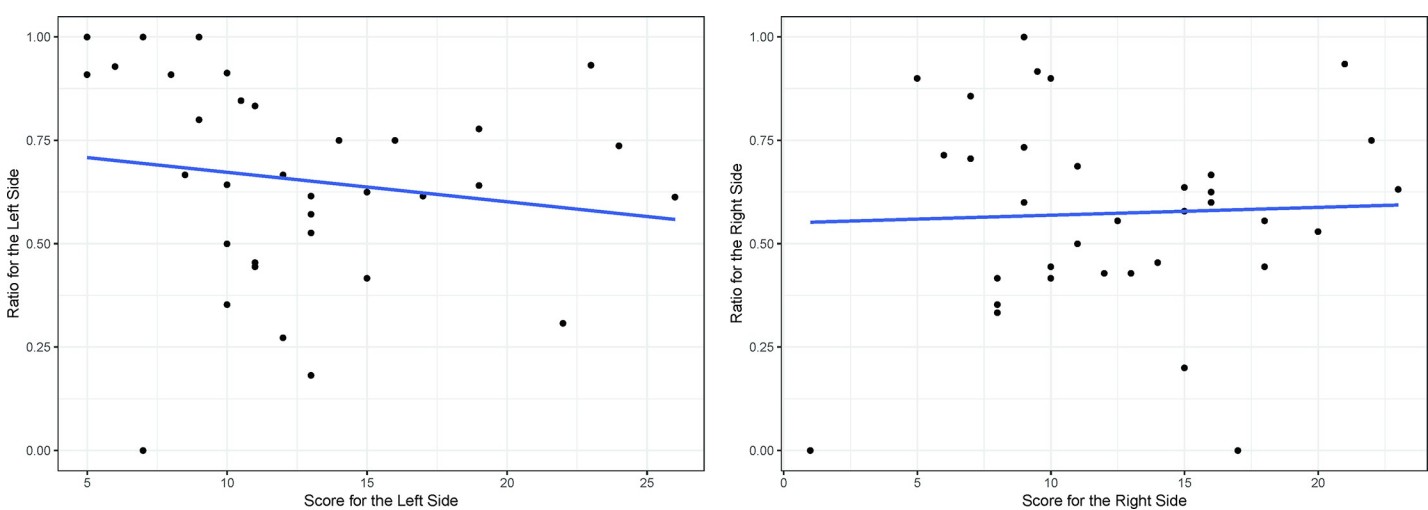

**Fig 4. The UPDRS part III DBS on/off ratios at 6 months.** Left(A) and Right(B). The UPDRS Part III DBS On/Off Ratios at 6 Months and the Preoperative Part III Score (Medication Off) were not correlated in each side.

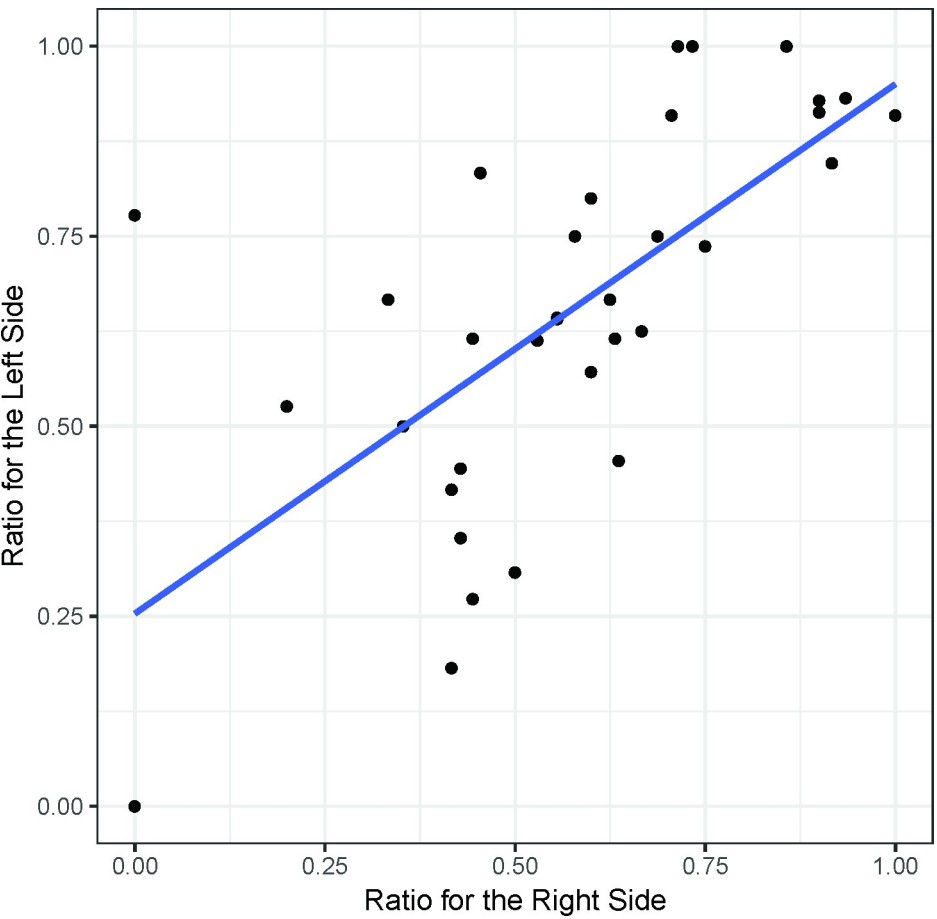

**Fig 5. The UPDRS part III DBS on/off ratios at 6 months (Medication Off).** The UPDRS Part III DBS On/Off Ratios at 6 months (Medication Off) show a linear correlation between the right and left sides, probably because a well-inserted electrode also has a positive effect on ipsilateral motor function as well as contralateral motor function.

One study showed that the high-frequency (500–2,000 Hz) background reflects the location of the STN [4]. Surprisingly, the results of single-task learning in the 50–500 Hz wavelet were the best. The 50–500 Hz wavelet appears to be the most consistent feature for the CNN.

## Multitask learning

The preoperative UPDRS part III score does not affect the DBS on/off ratio at 6 months after the operation (Fig 4A and 4B). In other words, no significant correlation was found between the preoperative condition and postoperative improvement, which is likely because the position of the electrode affected the clinical outcome. In this study, the left and right ratios were correlated with each other at 6 months (Fig 5), probably because a well-inserted electrode also has a positive effect on ipsilateral motor function. Unilateral STN DBS can improve the bilateral UPDRS motor scores of PD patients. The ratio of ipsilateral side symptom improvement to the contralateral side is approximately 1.63–2.09:1 according to previous research [23–25]. Although these studies did not show DBS off scores at 6 months, similar results can be deduced because a large difference between the preoperative score and the DBS off scores at 6 months was not found. To overcome the instability and inaccuracy of the single-task learning model, multitask learning using ipsilateral symptom data would be a good option.

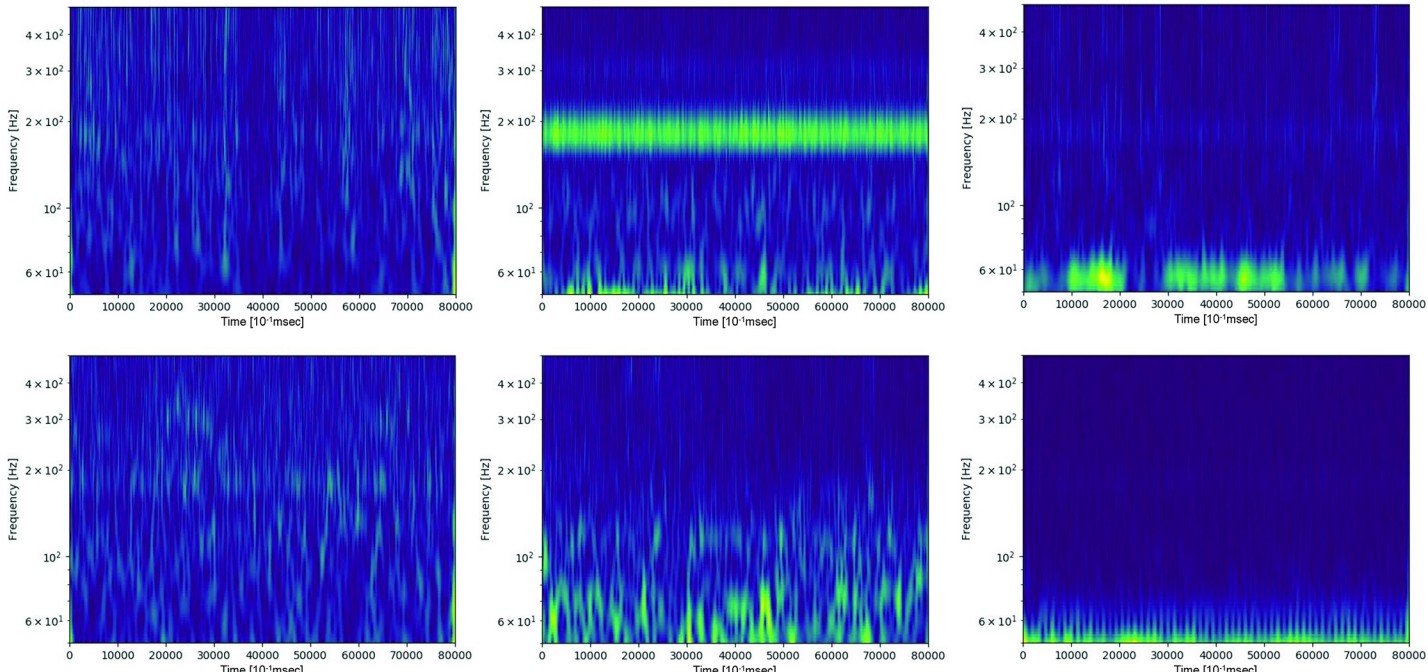

**Fig 6.** Examples of wavelets for the good-(A-C) and moderate (D-F)-response groups. No specific differences were observed between the two groups.

The maximal accuracy of over 70% occurred for the right-to-left loss ratio of 3–6:1. STN DBS has a bilateral effect, but it has a greater effect on the contralateral side, which is consistent with other papers that reported clinical outcomes in unilateral DBS [23–25]. Multitask learning is more stable than single-task learning because the bilateral effect is considered. Moreover, multitask learning itself can prevent overfitting. Considering this trait, the maximal accuracy being higher in the model that refers more to the contralateral score is meaningful.

### Expected clinical relevance

In our research, the STN is where the typical firing pattern is visually observed in the MER and where the final lead position is determined. In awake DBS, the results can be checked by macrostimulation of the selected target, but this verification is not possible in this study because asleep DBS was applied. Being able to predict the clinical outcome of DBS from a single MER would be helpful, particularly when a typical firing pattern is not visible. The method of manually determining the lead position is different, but it can be used as a common reference. Recently, published articles have predicted the clinical outcome from the electrode coordinates using deep learning [26]. However, no articles have used deep learning and the time information from MER data to measure the clinical outcome.

### MER under general anesthesia

While propofol anesthesia reduces the firing rate of some basal ganglia [27], no significant difference was found in firing rate with propofol and fentanyl sedation compared with local anesthesia [13]. MER could be performed properly without affecting the surgical outcome only when remifentanil dosing was stopped and the dose of propofol was carefully monitored [28–30]. However, the spontaneous firing patterns of STN and substantia nigra reticulate neurons remain similar to those of LA [30, 31]. No significant differences were found between the GA

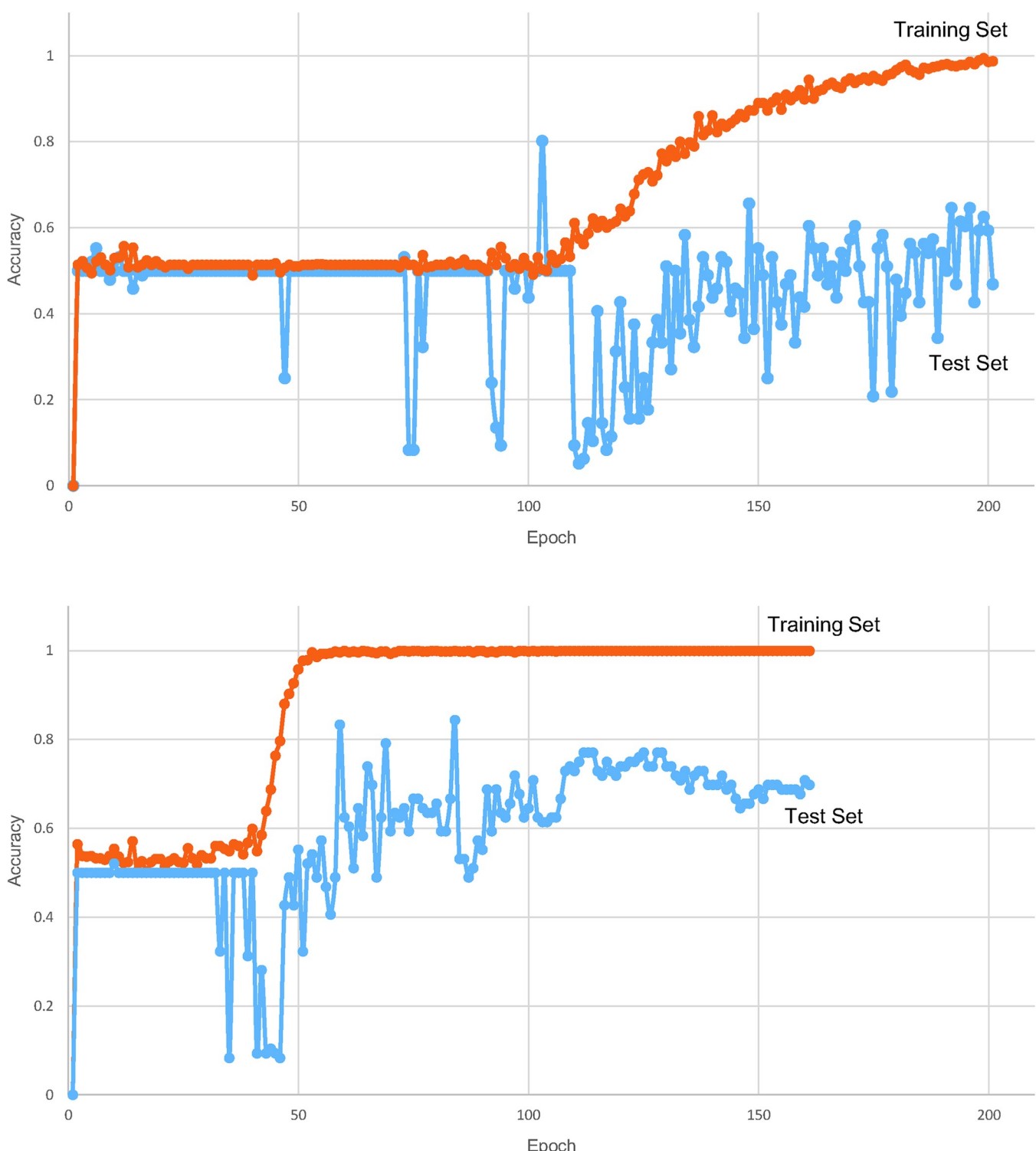

**Fig 7.** Accuracy of Single-task (A) Learning and Multitask (B) Learning by Learning Epoch. As the model repeats epochs, the results become unstable in single-task learning. The stability of the model with multitask learning was substantially improved over the model stability with single-task learning.

**Table 2. Maximal accuracy according to the right-to-left loss ratio.**

| Loss Ratio (Right:Left) | 1:5 | 1:4 | 1:3 | 1:2 | 2:3 | 1:1 | 3:2 | 2:1 | |
|---|---|---|---|---|---|---|---|---|---|
| Max. Accuracy (%) | 65.63 | 59.38 | 67.71 | 68.75 | 64.58 | 64.58 | 66.67 | 65.63 | |
| Loss Ratio (Right:Left) | 5:2 | 3:1 | 4:1 | 5:1 | 6:1 | 7:1 | 8:1 | 9:1 | 10:1 |
| Max. Accuracy (%) | 66.67 | 71.88 | 77.08 | 80.21 | 80.21 | 77.08 | 67.71 | 67.71 | 67.71 |

The maximal accuracy (80.21%) occurred when the right-to-left loss ratio was 5:1 and 6:1.

and LA groups for the MER trajectories, recorded STN-depths, pre- and postoperative coordinates, and overall incidences of stimulation side effects [32].

## Limitations

Left DBS was performed first, and right DBS was performed on the same day. The MER data from the right side were excluded because they were biased due to left-side DBS insertion.

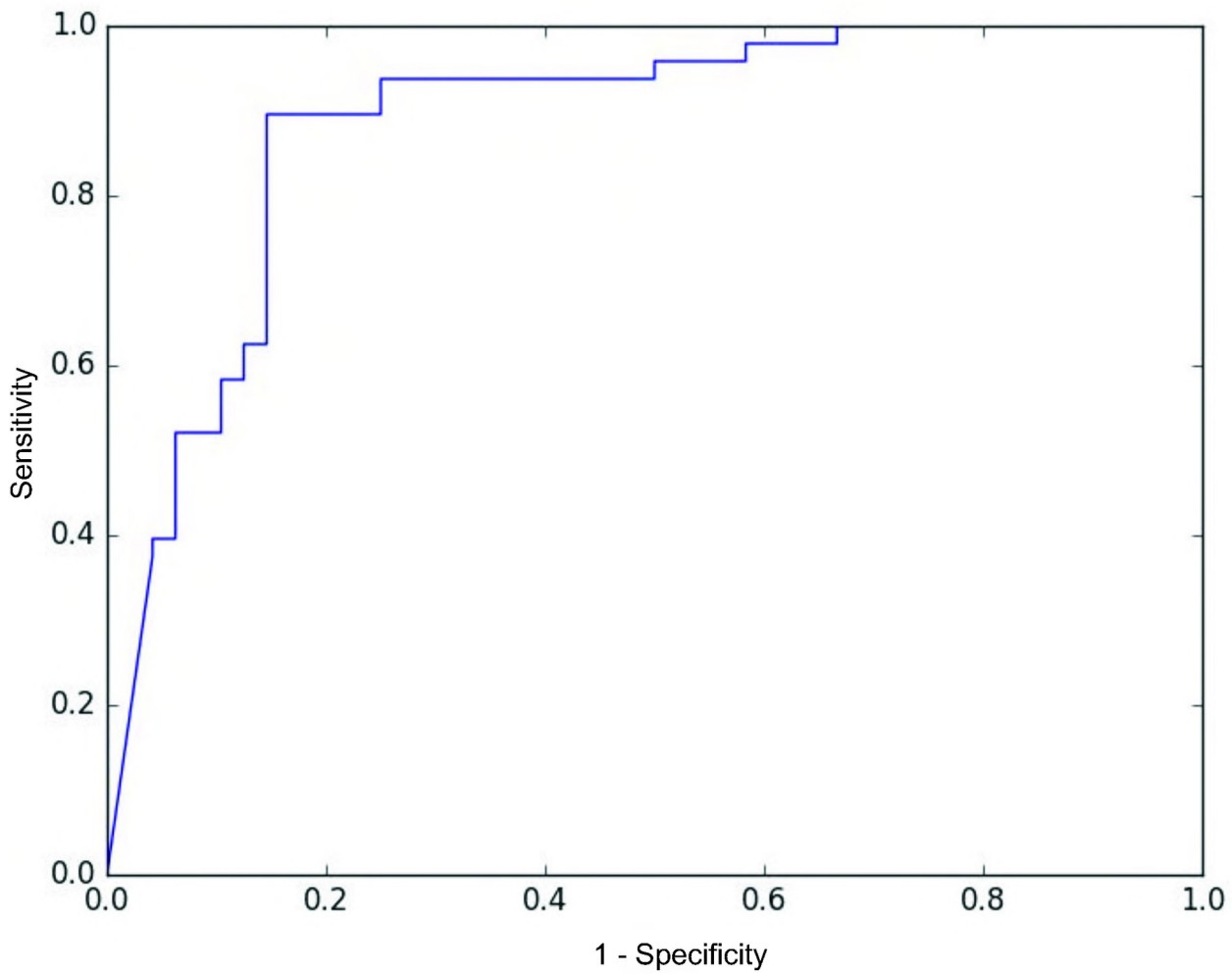

**Fig 8. Receiver operative characteristic curve.** The area under the curve (AUC) was 0.88.

Furthermore, only asleep DBS with the same anesthetic was included to reduce the anesthesia and awake biases, such as patient movements or snoring. Future studies will include all of the excluded data, and we will explore how to input the left and right MER signals into the learning algorithm at the same time. Because we used the data from 6 months after the surgery, we were unable to investigate surgical and medical problems, such as infections and disease progression. Cerebrospinal fluid (CSF) leakage or head positions can cause brain shifting during surgery, but we ignored this effect. Brain shifting can change the exact location of a lead at 6 months after surgery [33]. We expect to obtain better results by finding a technique to reflect each brain shift. As the three inputs were inevitably given the one output value, in future studies we will obtain better results by labeling each at a 1:1 ratio or by adding information about the spatial relationships. As the sampling rate of the signal was large and few well-known features about MER are available, we could not have a better machine learning control group. A personalized model was considered, but universal training can predict the MER signal without using the previous signal. Various optimizers were adopted for the CNN to converge, but the validation accuracy did not increase in the control group.

## Conclusion

Clinical improvements in PD patients who underwent bilateral STN DBS can be predicted from MER using multitask deep learning. The MER signal itself and the clinical outcome were matched end-to-end. In particular, multitask learning was used to consider the bilateral effect of unilateral DBS. The results of the experiment with different ratios of the loss function show that 5:1 and 6:1 have the best maximal accuracy (80.21%). We had expected to be able to use machine learning to determine the differences that cannot be found in the conventional way. However, the result cannot be accepted because of various limitations. After correcting the limitations one by one and satisfying the results, we will be able to determine the hidden factor in the MER data. We believe that this finding may be helpful for determining the appropriate electrode location in new patients.

## Supporting information

**S1 Material.**
(DOCX)

## Acknowledgments

### Previous presentation

Oral Presentation at the 18th Biennial Meeting of the World Society for Stereotactic and Functional Neurosurgery (New York City, June 24–27, 2019)

## Author Contributions

**Conceptualization:** Sukkyu Sun, Sun Ha Paek.

**Data curation:** Yong Hoon Lim, Hye Ran Park, Jae Meen Lee, Kawngwoo Park, Beomseok Jeon, Sun Ha Paek.

**Formal analysis:** Sukkyu Sun, Yong Hoon Lim, Hee Chan Kim, Sun Ha Paek.

**Funding acquisition:** Sun Ha Paek.

**Investigation:** Kwang Hyon Park, Sukkyu Sun, Sun Ha Paek.

**Methodology:** Kwang Hyon Park, Sukkyu Sun, Sun Ha Paek.

**Resources:** Beomseok Jeon, Sun Ha Paek.

**Software:** Sukkyu Sun.

**Supervision:** Beomseok Jeon, Hee Chan Kim, Sun Ha Paek.

**Validation:** Sukkyu Sun.

**Writing – original draft:** Kwang Hyon Park, Sukkyu Sun, Yong Hoon Lim.

**Writing – review & editing:** Kwang Hyon Park, Sukkyu Sun, Hee-Pyoung Park, Hee Chan Kim, Sun Ha Paek.

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
