## [Decision Letter · Decision Letter 0]

18 Feb 2020

PONE-D-20-01259

Clinical outcome prediction from a deep learning-based microelectrode recording analysis in subthalamic deep brain stimulation for Parkinson's disease

PLOS ONE

Dear Dr. Paek,

Thank you for submitting your manuscript to PLOS ONE. After careful consideration, we feel that it has merit but does not fully meet PLOS ONE’s publication criteria as it currently stands. Therefore, we invite you to submit a revised version of the manuscript that addresses the points raised during the review process.

We would appreciate receiving your revised manuscript by Apr 03 2020 11:59PM. To enhance the reproducibility of your results, we recommend that if applicable you deposit your laboratory protocols in protocols.io, where a protocol can be assigned its own identifier (DOI) such that it can be cited independently in the future. For instructions see: http://journals.plos.org/plosone/s/submission-guidelines#loc-laboratory-protocols

We look forward to receiving your revised manuscript.

Kind regards,

Academic Editor

PLOS ONE

Additional Editor Comments (if provided):

Please reply the comments from the reviewers. Revise accordingly if feasible.

Journal Requirements:

Reviewers' comments:

Reviewer's Responses to Questions

**Comments to the Author**

1. Is the manuscript technically sound, and do the data support the conclusions?

Reviewer #1: Partly

Reviewer #2: Yes

2. Has the statistical analysis been performed appropriately and rigorously? 

Reviewer #1: I Don't Know

Reviewer #2: Yes

3. Have the authors made all data underlying the findings in their manuscript fully available?

Reviewer #1: Yes

Reviewer #2: No

4. Is the manuscript presented in an intelligible fashion and written in standard English?

Reviewer #1: No

Reviewer #2: Yes

5. Review Comments to the Author

Reviewer #1: The authors test the hypothesis that intraoperative microelectrode recordings (MER) can be predictive of outcomes (as measured by UPDRS) 6 months after implantation of bilateral deep brain stimulation (DBS) electrodes in Parkinsonian patients. MER has been routinely used for more accurate positioning of DBS electrodes in patients who are asleep during the implantation procedure (as in this study). However, using intraoperative MER to predict post-operative outcome is new and is an that is likely to have great significance, if successful.

However, the manuscript is very poorly written and it took a lot of effort and time even to understand the central hypothesis of this study. The author should state the central hypothesis of their study in the introduction section and explain briefly why it is important. The statement of goal(s) in the abstract lacks any rationale (why is it important?) other than saying that it has not been done before. With substantial revisions, this manuscript might be acceptable for publication. The manuscript is lacking in detail – for example, it goes from a generic statement about neural nets to some very specific detail about the structure of the network in a couple of sentences. And that is just one of the examples. Many crucial details are missing making it very difficult to understand and critique some of the key results, claims and conclusions. For instance, despite much searching of the article, I could not find information on when exactly MERs were being done (was it done subsequent pre-, during or post-operatively), were the MER leads removed prior to DBS macrostimulation, what do the leads look like, how do the signals typically look like (some sample traces will be important before it is being used for wavelet analysis). The lack of detail is unsatisfactory as a technical article; perhaps the authors should target a clinical journal with more focus on patient data, procedures and results that might find such sparse description of technical details more acceptable. I have a long list of sentences in this article that were unclear in meaning. At some point, I had to give up. The article could really use a good, professional proof-reader.

Some (not exhaustive) examples where the description lacked clarity:

Lines 153-156 - Which 3 inputs and which output? When were these measurements done? “when the lead stimulation depth contact …” unclear what the authors are trying to say here.

Lines 176-179 - which leads? stim leads or the MER recording leads? What are the inputs and outputs? “The recorded MER signal was selected from the contact position at 6 months after DBS.” What does this mean?

Line 194 – what “requirements”?

Line 195 - What is a “large-capacity” signal? Are the authors suggesting large bandwidth signal? Perhaps they are using unfamiliar jargon and should clarify.

Line 197 - time and frequency resolution of 8. What does this mean? Is it time resolution or frequency resolution? What are the units?

Line 205 – What does this sentence mean?

Lines 218 -219 - what are the 3 classes? what 2 classes? This doesn’t make sense.

Line 221 - contradicts line 177 which indicates a sampling rate of 48 kHz.

Are “labels” (line 227) and “classes” (line 218) the same? Very confusing.

Line 228 – repeat sentence.

Line 234 - What process? What task 1?

Line 241 onward – The authors need to find a better way to communicate the hardware descriptions.

Caption of Fig. 5 - “DBS On/Off ratio”?

This is the ratio of UPDRS III scores when DBS is on to when it is off. The caption should be fixed to make this clear.

Line 310 - What is “training loss”?

Line 312-313 – unclear sentence

The whole description of multi-task learning versus single task learning is unclear. The authors give a very generic description which is very hard to understand how it was applied in this study.

Lines 404-405 - This is very confusing. Was the MER used to predict outcome, which was then subsequently used to adjust the position of the electrode. Was the outcome prediction used to readjust the position of the stimulation electrode? If so, that introduces bias in the results.

Reviewer #2: Dear authors, I owe you a great respect to this pioneering work. However, the patient's safety is a critical issue in any neurosurgical intervention, particularly when implementing a relatively novel technique such as deep learning using MER signaling was applied. Although deep learning not directly related to postoperative morbidity, clear mentioning of the rate of postoperative hemorrhage (which can amount to 2%) and permanent neurologic deficit (approximately 1%) should have been made. As per experience, I can consider the lack of mention of these complications (although referred to in the limitations section) as an appropriate cause for rejection, firstly as they are related to the MER placement process and secondly because the patient safety should be a vital concern in a such an innovative study.

Other factors that needs revision but not rejection include the need to elucidate clearly the rationale behind using the off-medication state only in the inclusion criteria and the need to include an awake DBS MER learning model, in addition to inclusion of a broader age groups spectrum, including older age groups, patients with higher BMI and DM (as some papers showed an impact on the postoperative outcome like that of Fernandez and Abboud 10.1016/j.heliyon.2019.e01862)

6. PLOS authors have the option to publish the peer review history of their article (what does this mean?). If published, this will include your full peer review and any attached files.

Reviewer #1: No

Reviewer #2: Yes: MM

---

## [Author Response · Author response to Decision Letter 0]

9 Sep 2020

Reviewer #1

The authors test the hypothesis that intraoperative microelectrode recordings (MER) can be predictive of outcomes (as measured by UPDRS) 6 months after implantation of bilateral deep brain stimulation (DBS) electrodes in Parkinsonian patients. MER has been routinely used for more accurate positioning of DBS electrodes in patients who are asleep during the implantation procedure (as in this study). However, using intraoperative MER to predict post-operative outcome is new and is an that is likely to have great significance, if successful.

However, the manuscript is very poorly written and it took a lot of effort and time even to understand the central hypothesis of this study. The author should state the central hypothesis of their study in the introduction section and explain briefly why it is important. The statement of goal(s) in the abstract lacks any rationale (why is it important?) other than saying that it has not been done before. With substantial revisions, this manuscript might be acceptable for publication. 

 We modifies the last paragraph of introduction and objective part of abstract for explaining why this investigation is important.

The manuscript is lacking in detail – for example, it goes from a generic statement about neural nets to some very specific detail about the structure of the network in a couple of sentences. And that is just one of the examples. 

 We understand. We revised the article several points for reader to understand easily. 

Many crucial details are missing making it very difficult to understand and critique some of the key results, claims and conclusions. For instance, despite much searching of the article, I could not find information on when exactly MERs were being done (was it done subsequent pre-, during or post-operatively), were the MER leads removed prior to DBS macrostimulation, what do the leads look like, how do the signals typically look like (some sample traces will be important before it is being used for wavelet analysis). The lack of detail is unsatisfactory as a technical article; perhaps the authors should target a clinical journal with more focus on patient data, procedures and results that might find such sparse description of technical details more acceptable. I have a long list of sentences in this article that were unclear in meaning. At some point, I had to give up. The article could really use a good, professional proof-reader.

 We think that it is common typical procedure of DBS. Only the procedure or materials that are not in common are described in this article. MERs cannot be performed preoperatively or postoperatively. DBS macrostimulation cannot be performed without MER leads removal. All detailed procedures are written in previous articles.

12. Kim W, Song IH, Lim YH, Kim MR, Kim YE, Hwang JH, et al. Influence of propofol and fentanyl on deep brain stimulation of the subthalamic nucleus. J Korean Med Sci. 2014;29(9):1278-86.

13. Lee WW, Ehm G, Yang HJ, Song IH, Lim YH, Kim MR, et al. Bilateral Deep Brain Stimulation of the Subthalamic Nucleus under Sedation with Propofol and Fentanyl. PLoS One. 2016;11(3):e0152619.

Some (not exhaustive) examples where the description lacked clarity:

Lines 153-156 - Which 3 inputs and which output? When were these measurements done? “when the lead stimulation depth contact …” unclear what the authors are trying to say here.

 We changed the sentence as below 

“Since the lead contacts are 1.5 mm long and MER was performed at intervals of 0.5 mm, the three MER signals were inevitably matched to the same clinical outcome as input and output values in the deep learning model. When the lead stimulation depth contact was changed at 6 months after surgery, the MER signal of depth thought to have been stimulated was selected.”

Lines 176-179 - which leads? stim leads or the MER recording leads? What are the inputs and outputs? 

 It seems to be confusing because the words input and output are used in both field. For example, “output signal recorded from MER” is “input of deep learning algorithm” and “output value of deep learning algorithm” is clinical outcome. We changed several words to help understanding.

“The recorded MER signal was selected from the contact position at 6 months after DBS.” What does this mean?

 There are four contacts in a DBS electrode. We mainly put the second deepest contact at the target position, but there are cases where the stimulus position is changed during the follow up period. If the stimulation position was changed according to the symptoms, the signal to be put into the algorithm was decided accordingly.

Line 194 – what “requirements”?

 That means computer hardware system requirements, required memory of RAM(Random Access Memory) . We have revised the above-mentioned sentence. removed following line. “and the requirements are high”

Line 195 - What is a “large-capacity” signal? Are the authors suggesting large bandwidth signal? Perhaps they are using unfamiliar jargon and should clarify.

 We have revised the above-mentioned sentence. 

“a large capacity signal can be reduced to a small capacity signal” to “large sampling rate signal can be reduced to few kilobytes of signal”.

Line 197 - time and frequency resolution of 8. What does this mean? Is it time resolution or frequency resolution? What are the units? 

For continuous wavelet transformation ObsPy library was used. Original equation is from [Kristeková, Miriam, Jozef Kristek, and Peter Moczo. "Time-frequency misfit and goodness-of-fit criteria for quantitative comparison of time signals." Geophysical Journal International 178.2 (2009): 813-825.] continuous wavelet transform of signal s(t) is defined by

〖CWT〗_((a,b)) {s(t)}= 1/√(|a| ) ∫_(-∞)^∞▒〖s(t) Ѱ^* ((t-b)/a)dt〗

T being time, a is scale parameter, b is translational parameter and Ѱ is analysin wavelet.star denotes the conjugate function. The scale parameter a is inversely proportional to frequency f. A Morlet wavelet 

Ѱ(t)= π^((-1)/4) e^((iω_0 t)) e^((〖-t〗^2/2))

With ω_0=8 .

We have revised.

Line 205 – What does this sentence mean?

 We have revised the sentences to “Data set is shuffled to prevent one patient from including both train set and test set.”

Lines 218 -219 - what are the 3 classes? what 2 classes? This doesn’t make sense. Original structure of fully connected layer of VGG 16 was 4096 4096 1024, and in this study fully connected layer of original VGG16 modified to 120 120 16 2. 

 We totally agree with you. We have revised the above-mentioned sentence. “3” to “1024”

Line 221 - contradicts line 177 which indicates a sampling rate of 48 kHz.

The original signal with a sampling rate of 48kHz was resample to 20kHz. 

 We have revised the above-mentioned sentence. We removed following sentence “It is difficult to use an RNN to analyze MERs because the sampling rate is large (20 kHz) and the requirements are large. In a wavelet analysis, a large capacity signal can be reduced to a small capacity signal.”

Are “labels” (line 227) and “classes” (line 218) the same? Very confusing.

 We have revised the above-mentioned sentence. We changed “labels” to “classes” in the Multitask Learning paragraph.

Line 228 – repeat sentence.

 We removed following sentence “learning with two labels is more generalizable than training with one label.”

Line 234 - What process? What task 1?

 We removed “task 1” in the schematic diagram. But we forgot to change the sentence. It is now revised. 

Line 241 onward – The authors need to find a better way to communicate the hardware descriptions.

 We totally agree with the reviewer’s opinion and moving hardware and soft descriptions to supplementary material.

Caption of Fig. 5 - “DBS On/Off ratio”?

 This is the ratio of UPDRS III scores when DBS is on to when it is off. 

Line 310 - What is “training loss”?

 Training loss is meaning softmax cross entropy loss when training phase. We changed following sentence “training loss” to “softmax cross entropy loss”

Line 312-313 – unclear sentence

 We changed following sentence “However, as the model repeats epochs, the results become unstable (Fig 7A).” to “However the model repeated learning for each epoch, fluctuation occurred in validation accuracy at the end of training.”

The whole description of multi-task learning versus single task learning is unclear. The authors give a very generic description which is very hard to understand how it was applied in this study.

 We totally agree with the reviewer’s opinion that more information is needed for multi-task learning. We have now included sentences how we used multi-task learning in the “method section.” 

Lines 404-405 - This is very confusing. Was the MER used to predict outcome, which was then subsequently used to adjust the position of the electrode. Was the outcome prediction used to readjust the position of the stimulation electrode? If so, that introduces bias in the results.

 MER signal were analyzed via characteristics such as spikes and frequencies as usual. MER analysis with deep learning was not used to adjust the position of the electrode in this study. It is analyzed retrospectively. We deleted the statement of previous usual procedure. 

Reviewer #2

Dear authors, I owe you a great respect to this pioneering work. However, the patient's safety is a critical issue in any neurosurgical intervention, particularly when implementing a relatively novel technique such as deep learning using MER signaling was applied. Although deep learning not directly related to postoperative morbidity, clear mentioning of the rate of postoperative hemorrhage (which can amount to 2%) and permanent neurologic deficit (approximately 1%) should have been made. As per experience, I can consider the lack of mention of these complications (although referred to in the limitations section) as an appropriate cause for rejection, firstly as they are related to the MER placement process and secondly because the patient safety should be a vital concern in a such an innovative study.

 We understand the problem. This is just beginning of the deep learning based analysis. We will consider other problems in the algorithm in the further study. 

Other factors that needs revision but not rejection include the need to elucidate clearly the rationale behind using the off-medication state only in the inclusion criteria and the need to include an awake DBS MER learning model, in addition to inclusion of a broader age groups spectrum, including older age groups, patients with higher BMI and DM (as some papers showed an impact on the postoperative outcome like that of Fernandez and Abboud 10.1016/j.heliyon.2019.e01862)

 We perform only asleep STN DBS. It seems that it can be solved through collaboration with other centers. We understand the limitation of conditions not considered such as age, BMI, DM. However, we think it is of great significance to apply end-to-end deep learning and to present the beginning of a new analysis method that utilizes CNN and multitask learning in the signal analysis.

 

Reviewer #3

The authors report on the clinical prediction of the clinical outcome of DBS using deep learning. While the main idea is interesting, the study requires substantial revision. The main criticism is due to the fact the authors did not experiment with different machine /deep learning algorithms to find an algorithm that performs well. In conclusion, the authors mention the following 

“We expected to be able to use machine learning to determine the differences that cannot be found in a conventional way. However, it is not possible to accept the result because of various limitations. After correcting the limitations one-by-one and satisfying the results, we will be able to determine the hidden factor in the MER data.” 

This statement which aims to summarize the finding, shows that the study does not add much to the reader, because finding the right ML/DL tool and/or performing extensive parameter search to obtain the right algorithm is the main task when one hopes to employ DL/ML. By design, the ML/DL algorithms require fine-tuning to achieve the best outcome. Hence I recommend that the authors attempt to either modify the convolutional Neural network they use or look into other types of ML/DL tools that provide a satisfactory prediction. 

 We understand the limitation. However, we think it is of great significance to apply end-to-end deep learning and to present the beginning of a new analysis method that utilizes CNN and multitask learning in the signal analysis.

I have to add that the presentation and the flow of the text needs improvement as well. The paragraphs do not follow well, and at times the reader needs to think hard to understand the transition. 

In the following, I also have some more comments which I hope the author can take into account in the next version:

 The title of the paper needs to be modified. As of now, one struggle to read it because of long compound nouns “deep learning-based microelectrode recording analysis” and “subthalamic deep brain stimulation” I recommend something as follows: 

Analysis of microelectrode recordings from Deep brain stimulation using Deep learning or Leveraging deep learning for analysis of microelectrode recordings from Deep brain stimulation. Long nouns statements make the title hard to read and are not recommended by the English language educators (For more see “Williams, Joseph M., and Joseph Bizup. Lessons in clarity and grace. Pearson, 2014.”) 

We agree with you. We will change the title to “Clinical outcome prediction from analysis of microelectrode recordings using deep learning in subthalamic deep brain stimulation for Parkinson`s disease” 

 Do not define any acronyms in the abstract. The reader won’t remember them and one has to go back many times. 

We understand what you are worrying. However, if we don't use acronyms, we are concerned that the abstract will be too long to understand. For most stereotactic and functional neurosurgeons, those acronyms such as PD, DBS, MER, UPDRS are commonly used. Some other acronyms such as VGG, AUC, ROC were used to help understanding, because they are familiar then the full name. 

 The abstract needs to be a coherent text, I recommend combining all 4 parts. One can avoid stating the method in detail; just mention the goal of the study and state the outcomes. The purpose of an abstract is to spark the reader’s interest in the story and motivate them to continue reading the whole paper. An easy-to-follow and compelling writing style and story-telling is needed in the abstract. Also, diving into the details of the method in the abstract may not be the best. 

We agree with your purpose. We made some changes to abstract to make it easier for readers to understand and find interest. It is impossible to delete more basic information from the method, because the novel point of this article is the methods.

 Citation for relevant literature is missing in the introduction; make sure please review the relevant DBS literature in the introduction. The authors need to guide the reader ensuring that the reader can understand the significance of the issue by providing a coherent and cohesion text.

a. The text in the introduction jumps between topics and makes it rather hard to follow. For instance in line 12-127 the authors suddenly jump from discussing PD to Convolutional Neural Networks. This qualifies as a sharp and unexpected turn which leaves the readers baffled for a second. See the highlighted text below please 

“...no studies have deep learning-based analyze MER fsignals with the clinical outcomes of patients with advanced PD after STN DBS. Convolutional neural network (CNN) models can process signals using imaging processing methods” 

The authors need to make a case for the specific tool and provide smooth transitions. Also when discussing CNN, one has to show why it is the best tool for this task? As we can see later on the text CNN is not the best tool! 

We totally agree with you. That sentence was duplicated in the wrong paragraph by mistake. The next paragraph is the smooth transition you mentioned. Thank you. 

 At the end of the introduction please outline the study so the reader knows is coming next.

We revised last paragraph. 

 Only off-medication status was included in this study. It would be informative to consider both on and off medication. 

We understand your opinion. We thought off-medication status was the most important point of the outcome of the surgery. But we are now looking for the way to consider other conditions such as age, BMI, DM, postoperative events. However, we think it is of great significance to apply end-to-end deep learning and to present the beginning of a new analysis method that utilizes CNN and multitask learning in the signal analysis.

 Do the authors did any sort control group to see whether CNN performs better or any different on the data from that group? If not, it would be useful to have a control group. Also before using CNN on the MER data from the patients, do we have know indication that CNN will perform well on the MER data? Simply assuming CNN will work on this data because it has worked on other data is not the best argument. 

We totally agree with the reviewer’s comment on control group. To feed the raw signal to the ML algorithm, the sampling rate of the signal is large and there are few well known features about MER. So we used image based CNN method. We have added sentences to limitation session.

 “Since the sampling rate of the signal is large and there are few well known features about MER. So we could not have a better machine learning control group.”

 The authors train CNN using data from four patients; and used this to predict the outcome for other patients; This could be the source of possible error and the reason for the poor performance in the prediction. Maybe one can use data from each patient to train a patient-specific CNN. Why would a CNN trained with data from four patients should be able to predict the outcome for all other patients? 

We totally agree with the reviewer’s comment on performance prediction. CNN was trained with thirty patients and test was done by four patients. Four patient was chosen randomly. And we have considered about personalized model but universal training can predict MER signal without using the previous signal. We have added sentences to limitation session.

“Personalized model was considered but universal training can predict MER signal without using the previous signal”

 In the Deep learning section, one has to present more than a general knowledge and dive deep into why the DL is useful, and talk in depth about the tool used, why it is the best, and the reason behind the choice of the parameters for the DL algorithm. From the text one might think RNN and CNN are the only artificial neural networks. 

We totally agree with the reviewer’s comment general knowledge about deep learning. We added several sentences to methods section.

“CNN extracts features from images using convolution and pooling layers, extracted features were classified with fully connected layers”

 I recommend moving the software/hardware and and the statistical tools specification to the supplementary material, maybe one can include a code; most certainly a new section is not required for each specification. 

We totally agree with the reviewer’s comment about software/hardware and moved to supplementary material. We moved following hardware and software specifications to Supplementary material.

 The results section needs to be extended and provide a detailed explanation of the results; simply putting the figures; is not enough . 

We revised some paragraphs of the result section. 

 In lines 361-362 references are needed for this statement “which is consistent with other papers that reported clinical outcomes in unilateral DBS” 

We totally agree with you. It is revised. 

 In Fig 13 where the authors present the accuracy for test and training data, the poor performance in the test data is an indication of the poor learning. The accuracy of around 0.5, is not statistifactoty and one has to One can do an extensive parameter search, and/or try different algorithms to and find something suitable for the given problem. 

We totally agree with the reviewer’s comment about optimization of control group. Various optimizers were adopted to converge the CNN, but the validation accuracy did not increase. We added several sentences to limitation section.

 “Various optimizers were adopted to converge the CNN, but the validation accuracy did not increase in control group.”

 The figures need to be combined so the reader will look at one figure instead of many that present similar data, also teh captions could be extended to make sure the reader can understand what is presented in the figure without having to go back to the text.

We agree with you. We revised some captions.

---

## [Decision Letter · Decision Letter 1]

25 Sep 2020

PONE-D-20-01259R1

Clinical outcome prediction from analysis of microelectrode recordings using deep learning in subthalamic deep brain stimulation for Parkinson`s disease

PLOS ONE

Dear Dr. Paek,

Thank you for submitting your manuscript to PLOS ONE. After careful consideration, we feel that it has merit but does not fully meet PLOS ONE’s publication criteria as it currently stands. Therefore, we invite you to submit a revised version of the manuscript that addresses the points raised during the review process.

Please revise and address the reviewers' concerns.

We look forward to receiving your revised manuscript.

Kind regards,

Robert Jeenchen Chen MD MPH, MD, MPH

Academic Editor

PLOS ONE

Reviewers' comments:

Reviewer's Responses to Questions

**Comments to the Author**

1. If the authors have adequately addressed your comments raised in a previous round of review and you feel that this manuscript is now acceptable for publication, you may indicate that here to bypass the “Comments to the Author” section, enter your conflict of interest statement in the “Confidential to Editor” section, and submit your "Accept" recommendation.

Reviewer #2: All comments have been addressed

Reviewer #3: (No Response)

2. Is the manuscript technically sound, and do the data support the conclusions?

Reviewer #2: Yes

Reviewer #3: Yes

3. Has the statistical analysis been performed appropriately and rigorously? 

Reviewer #2: Yes

Reviewer #3: Yes

4. Have the authors made all data underlying the findings in their manuscript fully available?

Reviewer #2: Yes

Reviewer #3: Yes

5. Is the manuscript presented in an intelligible fashion and written in standard English?

Reviewer #2: Yes

Reviewer #3: No

6. Review Comments to the Author

Reviewer #2: I have no current comments except for the inclusion of the safety issues in the future works, which I had previously clarified in the previous review.

Reviewer #3: This is an interesting and novel proposal to predict postsurgical results. In this retrospective study you have recollected a valuable dataset and the statistical analysis has been run appropriately. In general terms, the written English, the prose of the manuscript, and the medical terminology leaves much room for improvement and at times, perhaps, language translation errors play a roll in this manuscript. I highly recommend a complete review of the prose and the medical terminology utilized to narrate the background, findings and results. In terms of the usage of numbers in their numerical format versus written format, I suggest developing a consistent and singular style throughout the manuscript. In my opinion, the narrative will reach the elegant and professional level require if a native English speaker scientist help to re-write this manuscript.

Please indicate the co-author Young Hoo Lim degree.

The scales numbers in figure#1 are illegible.

Abstract

A lot of time and some assumptions was necessary to understand the ideas that were intended to describe. There is a lack of connection between the correct engineering and clinical terminology. Likewise, no details that complete and concrete the evidence previously publish is mentioned across the whole abstract. I recommend consider the possibility to re-write it.

For example, in line 88 (the objective section) you start the paragraph saying what you did, is not clear what are the outcomes expected. You wrote: to better predict motor function improvement; motor function is a general term, are you referring to tremor? Rigidity? Bradykinesia? Postural instability? In line 90, you wrote: we could make the outcome better even under restriction. What is the outcome?

The methods section is very confusing; therefore, the results are unclear.

Introduction

In the introduction (line 122-123) you mentioned the definition and targeting of the STN has been improved due to advances in magnetic resonance imaging (MRI) techniques. Are you referring to the accuracy of in vivo identification and localization of the brain anatomical structures in behalf of the resolution improvements in MRI (3T or 7T)? If this is relevant as a background in this study, can you provide reference?

Also each line of the introduction require some adjusted like the above.

Dear author, in this point, I congratulate you for the innovation and the effort putting this date together. This is a valuable date and deserve an elegant and well explained presentation, respectfully, I suggest re-write the manuscript and re-submit to this journal again.

7. PLOS authors have the option to publish the peer review history of their article (what does this mean?). If published, this will include your full peer review and any attached files.

Reviewer #2: **Yes: **MM

Reviewer #3: **Yes: **Cinthya Aguero

---

## [Author Response · Author response to Decision Letter 1]

9 Nov 2020

Reviewer #2: I have no current comments except for the inclusion of the safety issues in the future works, which I had previously clarified in the previous review.

=> Thanks for your comment. This study was restrospective study. In future research, we will consider safety issues.

Reviewer #3: This is an interesting and novel proposal to predict postsurgical results. In this retrospective study you have recollected a valuable dataset and the statistical analysis has been run appropriately. In general terms, the written English, the prose of the manuscript, and the medical terminology leaves much room for improvement and at times, perhaps, language translation errors play a roll in this manuscript. I highly recommend a complete review of the prose and the medical terminology utilized to narrate the background, findings and results. In terms of the usage of numbers in their numerical format versus written format, I suggest developing a consistent and singular style throughout the manuscript. In my opinion, the narrative will reach the elegant and professional level require if a native English speaker scientist help to re-write this manuscript.

=> At the time of the first submission, I received English correction from a professional proofreading company. This revised version was now also received English correction. Attach the certificate as well.

=> We did complete review of prose and the medical terminology with the help of a professional translation correction company.

Please indicate the co-author Young Hoo Lim degree.

=> Thank you. It is revised.

The scales numbers in figure#1 are illegible.

=> Sorry, but the scale in the figure is not important, because it was just one example of the frequency ranges. And the scale is same as figure 6A-F

Abstract

A lot of time and some assumptions was necessary to understand the ideas that were intended to describe. There is a lack of connection between the correct engineering and clinical terminology. Likewise, no details that complete and concrete the evidence previously publish is mentioned across the whole abstract. I recommend consider the possibility to re-write it.

=> It is modified. We added some details. 

For example, in line 88 (the objective section) you start the paragraph saying what you did, is not clear what are the outcomes expected. You wrote: to better predict motor function improvement; motor function is a general term, are you referring to tremor? Rigidity? Bradykinesia? Postural instability? 

=> motor function means UPDRS part III scores including rigidity, bradykinesia, postural instability.

In line 90, you wrote: we could make the outcome better even under restriction. What is the outcome?

=> That means UPDRS part III score. It is modified. 

The methods section is very confusing; therefore, the results are unclear.

=> It is modified. We added some details. 

Introduction

In the introduction (line 122-123) you mentioned the definition and targeting of the STN has been improved due to advances in magnetic resonance imaging (MRI) techniques. Are you referring to the accuracy of in vivo identification and localization of the brain anatomical structures in behalf of the resolution improvements in MRI (3T or 7T)? If this is relevant as a background in this study, can you provide reference?

=> Yes. We added the reference.

Also each line of the introduction require some adjusted like the above.

---

## [Decision Letter · Decision Letter 2]

4 Dec 2020

Clinical outcome prediction from analysis of microelectrode recordings using deep learning in subthalamic deep brain stimulation for Parkinson`s disease

PONE-D-20-01259R2

Dear Dr. Paek,

We’re pleased to inform you that your manuscript has been judged scientifically suitable for publication and will be formally accepted for publication once it meets all outstanding technical requirements.

Kind regards,

Academic Editor

PLOS ONE

Additional Editor Comments (optional):

Reviewers' comments:

Reviewer's Responses to Questions

**Comments to the Author**

1. If the authors have adequately addressed your comments raised in a previous round of review and you feel that this manuscript is now acceptable for publication, you may indicate that here to bypass the “Comments to the Author” section, enter your conflict of interest statement in the “Confidential to Editor” section, and submit your "Accept" recommendation.

Reviewer #2: All comments have been addressed

Reviewer #3: All comments have been addressed

2. Is the manuscript technically sound, and do the data support the conclusions?

Reviewer #2: Yes

Reviewer #3: Yes

3. Has the statistical analysis been performed appropriately and rigorously? 

Reviewer #2: Yes

Reviewer #3: Yes

4. Have the authors made all data underlying the findings in their manuscript fully available?

Reviewer #2: Yes

Reviewer #3: Yes

5. Is the manuscript presented in an intelligible fashion and written in standard English?

Reviewer #2: Yes

Reviewer #3: Yes

6. Review Comments to the Author

Reviewer #2: I appreciate your retrospective work and would like you to consider the safety details in the future prospective work.

Reviewer #3: This novel proposal is now very well written, my previous comments were addressed. My last recommendation will be switch the word "patients" for the word "participants". congratulations for a great job.

7. PLOS authors have the option to publish the peer review history of their article (what does this mean?). If published, this will include your full peer review and any attached files.

Reviewer #2: **Yes: **MM

Reviewer #3: **Yes: **Cinthya Aguero

---

## [Editor Report · Acceptance letter]

9 Dec 2020

PONE-D-20-01259R2 

Clinical outcome prediction from analysis of microelectrode recordings using deep learning in subthalamic deep brain stimulation for Parkinson`s disease 

Dear Dr. Paek:

I'm pleased to inform you that your manuscript has been deemed suitable for publication in PLOS ONE. Congratulations! Your manuscript is now with our production department. 

Kind regards, 

on behalf of

Dr. Robert Jeenchen Chen 

Academic Editor

PLOS ONE